

# Active behaviour of terrestrial caterpillars on the water surface

Masakazu Hayashi[1] and Shinji Sugiura[2]

[1] Hoshizaki Green Foundation, Izumo, Shimane, Japan
[2] Graduate School of Agricultural Science, Kobe University, Kobe, Hyogo, Japan

## ABSTRACT

Most butterfly and moth larvae (Lepidoptera) are terrestrial. When terrestrial caterpillars accidentally fall into water, they may drown or be preyed upon by aquatic predators before they can safely reach land. However, how terrestrial caterpillars escape aquatic environments and predators remains unclear. In July 2018, we observed a terrestrial caterpillar actively moving forward on the surface of a pond in Japan until it successfully reached the shore. To further investigate this behaviour in terrestrial caterpillars, we experimentally placed larvae of 13 moth species (four families) on a water surface under laboratory and field conditions. All caterpillars floated. Larvae of seven species moved forward on the water surface, whereas those of six species did not. A total of two types of behaviour were observed; in *Dinumma deponens*, *Hypopyra vespertilio*, *Spirama retorta*, *Laelia coenosa*, *Lymantria dispar* (all Erebidae), and *Naranga aenescens* (Noctuidae), larvae swung their bodies rapidly from side to side to propel themselves along the water surface (*i.e.*, undulatory behaviour); in contrast, larvae of *Acosmetia biguttula* (Noctuidae) rapidly moved the abdomen (posterior segments) up and down for propulsion along the water surface (*i.e.*, flick behaviour). Although thoracic legs were not used for undulatory and flick behaviour, rapid movements of the abdomen were used to propel caterpillars on the water surface. We also observed that undulatory and flick behaviour on the water surface aided caterpillars in escaping aquatic predators under field conditions. In addition, we investigated the relationship between body size and undulatory behaviour on the water surface in the erebid *S. retorta* under laboratory conditions. The frequency and speed of forward movement on the water surface increased with body length. Together, these results show that the rapid movement of elongated bodies results in forward propulsion on the water surface, allowing some terrestrial caterpillars to avoid drowning or aquatic predators. We further suggested potential factors related to morphology, host plant habitat, and defensive behaviour that may have led to the acquisition of aquatic behaviour in terrestrial caterpillars.

## INTRODUCTION

Most terrestrial insects have not adapted to aquatic environments; for example, many terrestrial insect species only rarely escape from a water surface. However, terrestrial insects such as locusts, cockroaches, praying mantises, and ants can swim on a water surface using their legs (*Miller, 1972*; *Franklin, Jander & Ele, 1977*; *Pflüger & Burrows,*

Corresponding authors
Masakazu Hayashi,
hgf-haya@green-f.or.jp
Shinji Sugiura,
ssugiura@people.kobe-u.ac.jp

*1978*; *Graham et al., 1987*; *Bohn, Thornham & Federle, 2012*; *Yanoviak & Frederick, 2014*; *Gripshover, Yanoviak & Gora, 2018*). Swimming behaviour has been reported for the adult stages of terrestrial insects, but rarely for the immature stages.

The larvae of butterflies and moths (Lepidoptera) are predominantly terrestrial; however, approximately 0.5% of 157,000 known species are aquatic at the larval stage (*van Nieukerken et al., 2011*; *Pabis, 2018*). When terrestrial caterpillars accidentally fall into water, they can drown or be preyed upon by aquatic predators such as fish before they can safely reach land (*Gustafsson, Greenberg & Bergman, 2014*; *Iguchi et al., 2004*). Some caterpillars (*i.e.*, aquatic species) exhibit behavioural adaptations to aquatic environments and predators to avoid these risks (*Pabis, 2018*), but the behavioural responses of terrestrial caterpillars to aquatic environments remain unclear.

On July 20, 2018, we observed a terrestrial caterpillar of *Dinumma deponens* Walker (Lepidoptera: Erebidae) moving forward on the water surface of a pond in Unnan, Shimane, Japan. The caterpillar undulated from side to side, propelling itself forward on the water surface; it was able to successfully reach the shore (Fig. 1A). The caterpillar may have accidentally fallen into the pond because *D. deponens* larvae feed on leaves of the tree species *Albizia julibrissin* Durazz. (Fabaceae), which commonly grows along the edges of wetlands (*Kishida, 2011*). We placed the same caterpillar on the water surface again and observed the same behaviour (Fig. 1B; Video S1). This active behaviour on the water surface appeared to avoid drowning and aquatic predators (*i.e.*, water striders; Fig. 1B; Video S1).

To investigate the water surface behaviour in terrestrial caterpillars further, we experimentally placed the larvae of 13 moth species (belonging to four families), including *D. deponens*, onto a water surface and observed their behaviour under laboratory and field conditions. In addition, we experimentally investigated the relationship between caterpillar body size and aquatic behaviour to clarify how body size can influence propulsive power in water.

## MATERIALS AND METHODS

To test whether terrestrial caterpillars can move forward on the water surface, we experimentally placed the larvae of 13 moth species (from four families) on a water surface and observed their behaviour under laboratory and field conditions (Table 1). We collected 52 larvae from eight plant species from June 2019 to July 2019 in Shimane Prefecture and in June 2020 in Hyogo Prefecture, Japan. We carefully placed each caterpillar ($n = 49$) on the water surface in a plastic vessel (390 mm × 265 mm × 65 mm) containing 2 L of water (20 mm depth, 25 °C) under well-lit conditions, with an air temperature of 25 °C. We also placed the larvae of three species, *Hypopyra vespertilio* (Fabricius) (Erebidae), *Acosmetia biguttula* (Motschulsky) (Noctuidae), and *Theretra oldenlandiae* (Fabricius) (Sphingidae), on the surfaces of ponds in Shimane Prefecture. During each 2-min observation period, we investigated whether the larvae (1) remained at the water surface (supported by water tension) and (2) moved forward on the water surface. To examine the possible origins of aquatic behaviour, we also observed how caterpillars of each species walk on twigs or leaves (*i.e.*, inching or crawling; *van Griethuijsen & Trimmer, 2014*;

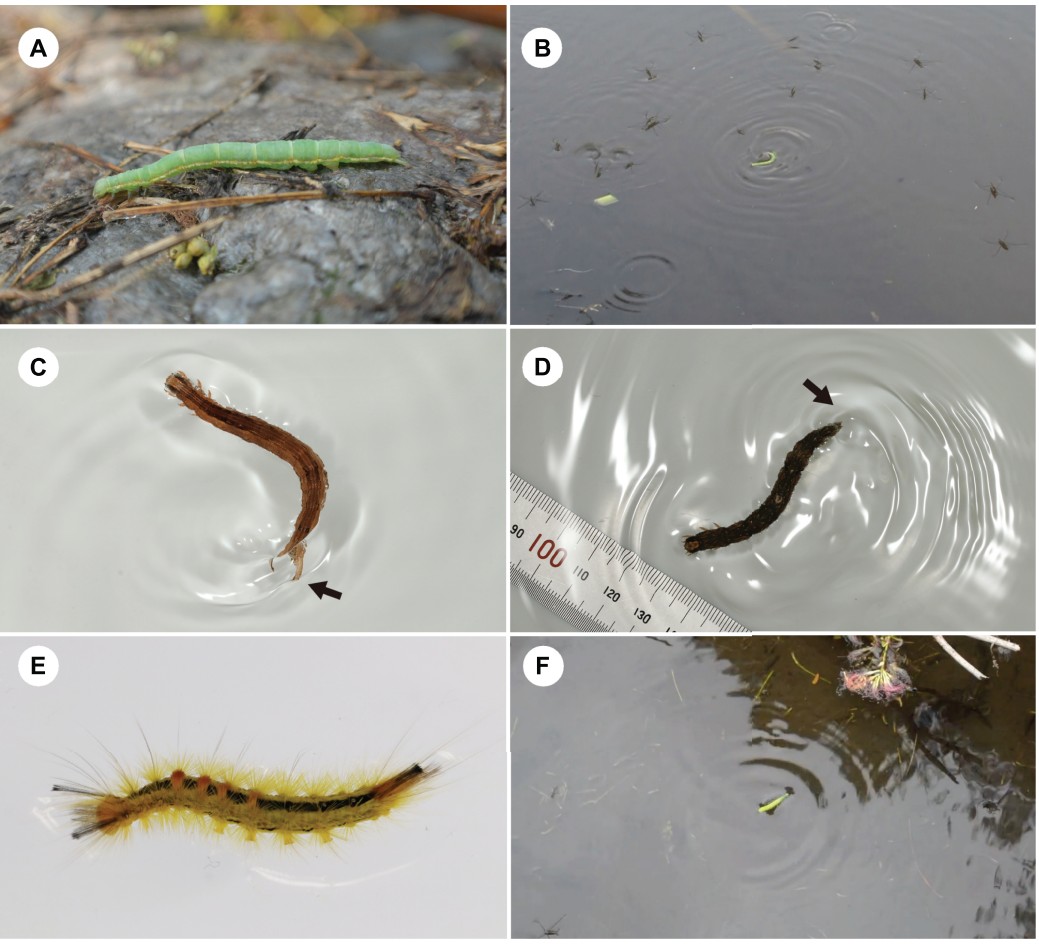

**Figure 1** **Behaviour of terrestrial caterpillars on the water surface.** (A) *Dinumma deponens* (Erebidae). (B) *Dinumma deponens* moving forward on a pond surface. (C) Undulatory behaviour in *Spirama retorta* (Erebidae). (D) Undulatory behaviour in *Hypopyra vespertilio* (Erebidae). (E) Undulatory behaviour in *Laelia coenosa* (Erebidae). (F) Flick behaviour in *Acosmetia biguttula* (Noctuidae). Arrows indicate anal prolegs. Photo credits: (A–D, F) M. Hayashi, (E) S. Sugiura.

Table 1). We identified each caterpillar based on their morphological characteristics (*Sugi, 1987*; *Yasuda, 2010*, *2012*, *2014*; *Suzuki et al., 2018*), and raised some larvae to the adult stage to confirm their identity (*Kishida, 2011*).

In caterpillars, various types of behaviour such as anti-predator defences are closely related to body size (*Sugiura & Yamazaki, 2014*; *Hossie et al., 2015*; *Sugiura, 2020*; *Sugiura et al., 2020*). To clarify how caterpillar size can influence propulsive power in water, we experimentally investigated the relationship between body size and water surface behaviour in the erebid *Spirama retorta* (Clerck) (Erebidae). We reared *S. retorta* larvae from the eggs of two females on *Albizia julibrissin* leaves under laboratory conditions (26–29 °C). *Spirama retorta* passes through seven larval instars before pupation (Table 2). We measured the body weight of each larva to the nearest 1 mg using an electronic balance (CJ-620S; Shinko Denshi, Co., Ltd., Tokyo, Japan); we measured the body length and head capsule width to the nearest 0.01 mm using slide callipers or an ocular micrometre.

**Table 1 Behaviour of the caterpillars placed on water surfaces.**

| Family | Species | Instar[a] | Length (mm) | Host plant range | Plant species (sampling) | Habitat (sampling) | Walking locomotion | Behaviour on water[b] | Forward movement on water % (n) |
|---|---|---|---|---|---|---|---|---|---|
| Erebidae | *Hypopyra vespertilio* | M–L | 23–70 | Fabaceae | *Albizia julibrissin* | Lake bank | Inching | Undulatory | 100 (7/7)[c] |
| | *Spirama retorta* | M–L | 8–42 | Fabaceae | *Albizia julibrissin* | Lake bank | Inching | Undulatory | 100 (3/3) |
| | *Dinumma deponens* | M–L | 20–32 | *Albizia julibrissin* | *Albizia julibrissin* | Lake bank | Inching | Undulatory | 33 (1/3) |
| | *Laelia coenosa* | L | 22–34 | Poaceae, Cyperaceae, Typhaceae | *Typha latifolia* | Pondside | Crawling | Undulatory | 100 (6/6) |
| | *Lymantria dispar* | L | 33–54 | Many families | *Cerasus × yedoensis* | Urban area | Crawling | Undulatory | 30 (3/10) |
| Noctuidae | *Xanthodes transversa* | M–L | 25–42 | Malvaceae | *Hibiscus mutabilis* | Garden | Inching | – | 0 (0/2) |
| | *Acosmetia biguttula* | M–L | 20–38 | *Bidens* | *Bidens frondosa* | Pondside | Crawling | Flick | 100 (6/6)[c] |
| | *Naranga aenescens* | M–L | 13–24 | Poaceae | *Pseudoraphis sordida* | Paddy field | Inching | Undulatory | 100 (4/4) |
| | *Sarcopolia illoba* | E–M | 19–34 | Many families | *Albizia julibrissin* | Lake bank | Crawling | – | 0 (0/3) |
| | *Britha inambitiosa* | M–L | 13–20 | *Pterostyrax hispidus* | *Pterostyrax hispidus* | Streamside | Inching | – | 0 (0/3) |
| Geometridae | *Chiasmia defixaria* | M–L | 20–30 | *Albizia julibrissin* | *Albizia julibrissin* | Lake bank | Inching | – | 0 (0/3) |
| | *Ectropis excellens* | L | 30 | Many families | *Pterostyrax hispidus* | Streamside | Inching | – | 0 (0/1) |
| Sphingidae | *Theretra oldenlandiae* | E | 20 | Many families | *Causonis japonica* | Garden | Crawling | – | 0 (0/1)[c] |

Notes:
[a] Instar: E, early instar; M, middle instar; L, late instar.
[b] Caterpillar behaviour on the water surface: Undulatory, forward movement by undulating; Flick, forward movement by flicking; –, non-forward movement (floating).
[c] One larva of each species was observed on the water surface of a pond, while other larvae were observed under laboratory conditions.

**Table 2 Body size and forward movement on the water surface in *Spirama retorta* larvae.**

| Instar | Body weight (mg)[a] | Body length (mm)[a] | Head width (mm)[a] | Floating (%) | Forward movement (%) | n |
|---|---|---|---|---|---|---|
| First | 0.4 ± 0.2 | 6.1 ± 0.2 | 0.4 ± 0.0 | 100 | 0 | 10 |
| Second | 8.4 ± 1.1 | 14.3 ± 0.5 | 0.7 ± 0.0 | 100 | 0 | 10 |
| Third | 27.9 ± 2.0 | 22.3 ± 0.6 | 1.3 ± 0.0 | 100 | 40 | 10 |
| Fourth | 79.1 ± 5.7 | 29.2 ± 0.5 | 2.0 ± 0.0 | 100 | 70 | 10 |
| Fifth | 281.6 ± 21.0 | 44.4 ± 0.9 | 2.7 ± 0.0 | 100 | 100 | 10 |
| Sixth | 587.4 ± 47.6 | 54.8 ± 1.4 | 3.5 ± 0.1 | 100 | 100 | 10 |
| Seventh | 884.8 ± 72.3 | 61.1 ± 1.3 | 4.1 ± 0.0 | 100 | 100 | 10 |

Note:
[a] Values are mean ± SE.

We placed ten larvae per instar individually on the water surface in a plastic vessel (390 mm × 265 mm × 65 mm) with 2 L of water (20 mm depth) under well-lit conditions at 25 °C. We filmed the behaviour of the larvae (n = 70) using a video camera

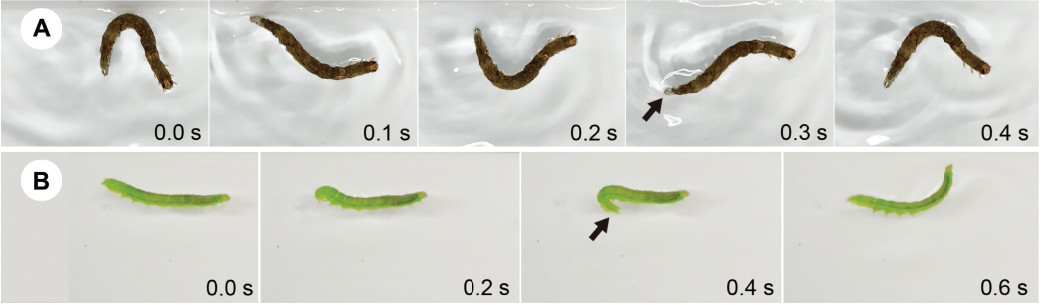

**Figure 2 A total of two types of caterpillar behaviour on the water surface.** (A) Temporal sequence of undulatory behaviour in *Hypopyra vespertilio*. (B) Temporal sequence of flick behaviour in *Acosmetia biguttula*. Arrows indicate anal prolegs. Photo credits: (A) S. Sugiura, (B) M. Hayashi.

(V2; Nikon, Tokyo, Japan). We played back the footage of the recorded behaviour using iMovie version 10.0.6 (Apple, Inc., Cupertino, CA, USA). During each 2-min observation period, we recorded (1) whether the larva remained at the water surface (supported by water tension), (2) whether the larva moved forward on the water surface, and (3) the distance (mm) travelled by the larva in 2 s.

To investigate the relationship between larval body length and aquatic behaviour in *S. retorta*, we ran a generalised linear model with a binomial error distribution and logit link function (*i.e.*, logistic regression). We used ten individuals per instar ($n = 70$) for the analysis. We used forward movement (1) or non-forward movement (0) on a water surface as the binary response variable; we regarded body length as a fixed factor. We also ran a generalised linear model with a Poisson error distribution and log link function (*i.e.*, Poisson regression) to investigate the relationship between body size and movement distance in *S. retorta*, analysing ten individuals per instar ($n = 70$). We used forward speed (mm/s) as the response variable; we regarded body length as a fixed factor. When the residual deviance was smaller (underdispersion) or larger (overdispersion) than the residual degrees of freedom, we used a quasi-binomial or quasi-Poisson error distribution, respectively, rather than a binomial or Poisson error distribution (*Sugiura & Sato, 2018*). We performed all analyses using R software version 3.5.2 (*R Core Team, 2019*).

## RESULTS

All caterpillars examined in this study floated (*i.e.*, remained at the water surface). Larvae from six of the 13 caterpillar species did not move forward on the water surface, whereas larvae from seven species (two families: Erebidae and Noctuidae) moved forward on the water surface (Table 1). A total of two types of behaviour were observed (Table 1): larvae of *Dinumma deponens*, *Hypopyra vespertilio*, *Spirama retorta*, *Laelia coenosa* (Hübner), *Lymantria dispar* (Linnaeus) (all Erebidae), and *Naranga aenescens* Moore (Noctuidae) swung their bodies side to side quickly to propel themselves on the water surface (*i.e.*, undulatory behaviour; Figs. 1C –1E, 2A ; Video S2), while larvae of *Acosmetia biguttula* (Noctuidae) moved the posterior segments of the abdomen up and down quickly to propel themselves on the water surface (*i.e.*, flick behaviour; Figs. 1F, 2B;

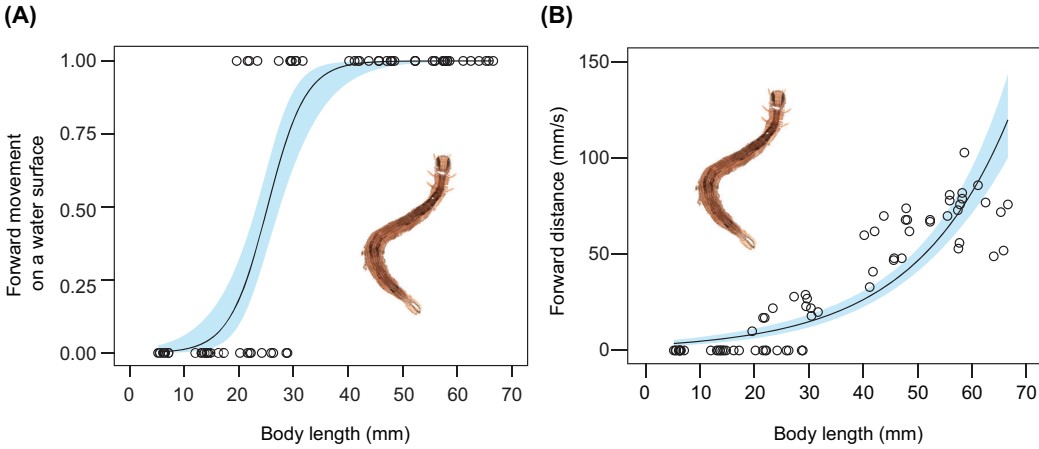

**Figure 3** Relationship between body size and behaviour in *Spirama retorta*. (A) Relationship between body length and frequency of undulatory behaviour (*n* = 70). (B) Relationship between body length and forward speed (mm/s) (*n* = 70). Lines and blue areas represent logistic regression lines and 95% confidence intervals derived from generalised linear models, respectively (Tables 3 and 4). Photo credit: M. Hayashi.                                

**Table 3** Relationship between body size and forward movement on the water surface in *Spirama retorta* larvae obtained using a generalised linear model.

| Response variable | Explanatory variable (fixed effect) | Coefficient estimate | SE | *t* value | *P* value |
|---|---|---|---|---|---|
| Forward movement on water[a] | Intercept | −7.21997 | 1.33807 | −5.396 | <0.0001 |
| | Caterpillar body length | 0.28593 | 0.05312 | 5.383 | <0.0001 |

**Note:**
[a] A quasi-binomial error distribution (rather than a binomial error distribution) was used because the residual deviance was smaller than the residual degrees of freedom (underdispersion).

**Table 4** Relationship between body size and forward distance (mm/s) on the water surface in *Spirama retorta* larvae obtained using a generalised linear model.

| Response variable | Explanatory variable (fixed effect) | Coefficient estimate | SE | *t* value | *P* value |
|---|---|---|---|---|---|
| Forward distance on water[a] | Intercept | 0.995874 | 0.233774 | 4.26 | <0.0001 |
| | Caterpillar body length | 0.056937 | 0.004376 | 13.01 | <0.0001 |

**Note:**
[a] A quasi-Poisson error distribution (rather than a Poisson error distribution) was used because the residual deviance was larger than the residual degrees of freedom (overdispersion).

Video S3). Thoracic legs were not used for undulatory or flick behaviour (Videos S2, S3). A total of one larva of *Acosmetia biguttula* was observed escaping from an aquatic predator (*Notonecta triguttata* Motschulsky) in a pond (Video S3).

The relationship between body size and undulatory behaviour in *S. retorta* was investigated under laboratory conditions. All larvae floated (Table 2). The frequency of forward movement on the water surface increased with body length (Fig. 3A; Tables 2 and 3): 0%, 0%, 40%, 70%, 100%, 100%, and 100% of the first, second, third, fourth, fifth, sixth, and seventh instars moved forward on the water surface, respectively (Table 2). Furthermore, the forward speed (mm/s) increased with body length (Fig. 3B; Table 4).

Larvae from eight of the 13 caterpillar species moved in a characteristic looping manner on leaves or stems (*i.e.*, inching; Table 1), whereas larvae from five species moved their abdomen up and down on land (*i.e.*, crawling; Table 1). When disturbed, larvae of *H. vespertilio*, *S. retorta*, and *D. deponens* were frequently observed to bend their bodies violently from side to side.

## DISCUSSION

Active behaviour on/under the water surface has been reported in some aquatic and semi-aquatic caterpillars (*Welch, 1914*; *Mey & Speidel, 2008*; *Meneses et al., 2013*; *Coates & Abel, 2019*; *De-Freitas, De Agostini & Stefani, 2019*). The aquatic larvae of *Paracles klagesi* (Rothschild) (Erebidae: Arctiinae) and *Neoschoenobia testacealis* Hampson (Crambidae) move and feed under the water surface (*Nagasaki, 1992*; *Meneses et al., 2013*), and semi-aquatic larvae of moths such as *Bellura vulnifica* (Grote) (Noctuidae) and *Ostrinia penitalis* (Grote) (Crambidae) can move forward on the water surface (*Welch, 1914*; *Coates & Abel, 2019*). However, few studies have examined whether typically terrestrial caterpillars can swim on or under the water surface. In the present study, we observed the behaviour on water surfaces of 13 terrestrial caterpillar species from four families under laboratory and field conditions. Among these, seven species were observed to move forward on the water surface (Figs. 1 and 2; Table 1), although none broke through the surface tension. We also observed two types of behaviour on the water surface (undulatory and flick behaviour) in the caterpillars (Figs. 1 and 2; Table 1). The undulatory behaviour observed in this study was similar to anguilliform movement, which has been reported in slender-bodied animals such as eels, snakes, and centipedes (*Graham et al., 1987*; *Sfakiotakis, Lane & Davies, 1999*; *Yasui et al., 2019*). The frequency and speed of undulatory behaviour increased with body length in *Spirama retorta* larvae (Fig. 3; Tables 3 and 4). Directed movements on the water surface can help caterpillars to avoid aquatic predators (Video S1).

All of the terrestrial caterpillars used in the present study floated due to water surface tension. Some, but not all, of these floating caterpillars moved forward on the water surface (Table 1). Three factors may influence forward movement on the water surface in terrestrial caterpillars: (1) morphology, (2) host plant habitat, and (3) locomotive and defensive behaviour.

Caterpillars that exhibited forward movement on the water surface had distinct morphological traits such as relatively elongated bodies. In this study, long-bodied caterpillars were more capable of forward movement on the water surface than those with short bodies (Fig. 3A; Table 3). This relationship has been suggested to explain the behaviour of the semi-aquatic caterpillar species *Bellura vulnifica*, although its morphological traits were not quantified (*Welch, 1914*). In addition, long body setae may assist in floating on the water surface in hairy caterpillars, such as those of *Laelia coenosa* and *Lymantria dispar* (cf., *Meyer-Rochow, 2016*). However, these features certainly evolved for reasons other than aquatic behaviour, because long bodies, prolegs, and body hairs have other important functions in their terrestrial habitats, *e.g.*, they may be involved in natural enemy defence, maintaining their perch, and still others

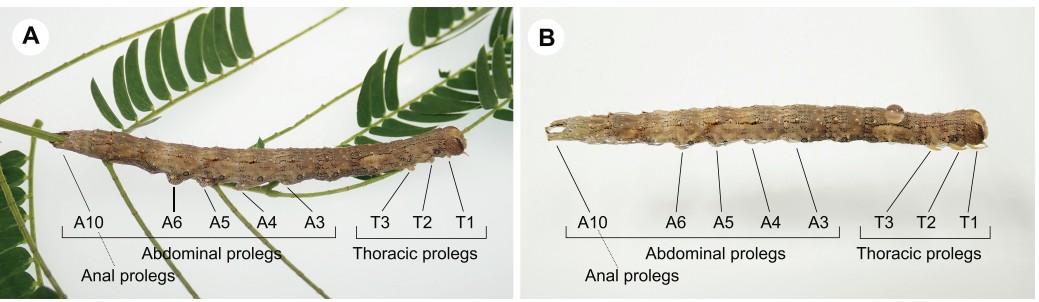

**Figure 4 Larval morphology of *Hypopyra vespertilio*.** (A) A larva on a host plant leaf. (B) A larva on the water surface. *Hypopyra vespertilio* larvae have three pairs of thoracic legs (T1–T3) and five pairs of abdominal prolegs (A3–A6 and A10). Photo credit: S. Sugiura.

(Fig. 4; *Skelhorn et al., 2010*; *van Griethuijsen & Trimmer, 2014*; *Sugiura & Yamazaki, 2014*).

Caterpillars use silk threads extruded from their spinnerets to disperse aerially (*Bell et al., 2005*) or drop from the host plant to escape natural enemies (*Sugiura & Yamazaki, 2006*). However, many mature microlepidopteran caterpillars descend from the host plant to the ground for pupation (*Sugi, 1987*). Caterpillars inhabiting host plants growing along watercourses may accidentally descend into open water. A total of six of the seven caterpillar species observed moving forward on the water surface in this study were collected from waterside plants such as *Albizia julibrissin* (Table 1).

Terrestrial behaviour can provide insight into the origins of aquatic behaviour in terrestrial caterpillars. Some caterpillars that undulated on the water surface typically locomote in a characteristic looping manner on leaves or stems (*i.e.*, inching; *van Griethuijsen & Trimmer, 2014*; Table 1). When disturbed, some larvae of *Hypopyra vespertilio*, *S. retorta*, and *Dinumma deponens* violently bent their bodies from side to side (*i.e.*, jerking, twisting, or thrashing behaviour; *Gross, 1993*; *Greeney, Dyer & Smilanich, 2012*). Undulating behaviour on the water surface may have originated from this defensive behaviour, rather than walking behaviour. Some caterpillars that exhibited flick behaviour typically moved their abdomen up and down to move on land (*i.e.*, crawling; *van Griethuijsen & Trimmer, 2014*; Table 1); the similarity of the flick behaviour and crawling motions suggests that flick behaviour on the water surface may have originated from crawling motion.

## CONCLUSIONS

Our results showed that some terrestrial caterpillars exhibited forward movement on the water surface to avoid drowning and aquatic predators (Table 1; Videos S1, S3). This behaviour is similar to the swimming behaviour reported in many aquatic and terrestrial animals (*Graham et al., 1987*; *Sfakiotakis, Lane & Davies, 1999*; *Yasui et al., 2019*). The behaviour found in this study was observed in only two of the four lepidopteran families tested: Erebidae and Noctuidae (Table 1). Our investigation was limited to four families, although the insect order Lepidoptera contains 133 recognised families (*Mitter, Davis & Cummings, 2017*). Thus, the behaviour observed in the erebids and noctuids

sampled in this study will probably be found in other lepidopteran families. Differences in aquatic skill among species should be investigated. Kinematic and anatomical studies would help elucidate the different mechanisms of aquatic behaviour in these and still other lepidopteran caterpillars.

## ACKNOWLEDGEMENTS

We thank the editor and reviewers for their helpful comments on an earlier version of the manuscript. We also thank K. Sakagami for aiding in moth identification and K. Okai for assistance with caterpillar sampling.

### Funding
The authors received no funding for this work.

### Competing Interests
The authors declare that they have no competing interests. Masakazu Hayashi is a researcher at the Hoshizaki Green Foundation.

### Author Contributions
- Masakazu Hayashi conceived and designed the experiments, performed the experiments, prepared figures and/or tables, authored or reviewed drafts of the paper, and approved the final draft.
- Shinji Sugiura conceived and designed the experiments, performed the experiments, analyzed the data, prepared figures and/or tables, authored or reviewed drafts of the paper, and approved the final draft.

### Data Availability
The data is available at Figshare: Hayashi, Masakazu; Sugiura, Shinji (2021): Data from: Active behaviour of terrestrial caterpillars on the water surface. figshare. Dataset. DOI 10.6084/m9.figshare.11891442.v1.

### Supplemental Information
Supplemental information for this article can be found online at http://dx.doi.org/10.7717/peerj.11971#supplemental-information.

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
