# Peer review of "Active behaviour of terrestrial caterpillars on the water surface"

_PeerJ, doi:10.7717/peerj.11971_

## Round 0.1 · original submission · Major Revisions

Dear Drs. Hayashi and Sugiura:

Thanks for submitting your manuscript to PeerJ. I have now received two independent reviews of your work, and as you will see, one reviewer recommended rejection, while another suggested a minor revision (though with many suggested changes). I am affording you the option of revising your manuscript according to both reviews but understand that your resubmission may be sent to at least one new reviewer for a fresh assessment (unless the reviewer recommending rejection is willing to re-review).

The reviewers raised many concerns about the manuscript. Please address all of these in your rebuttal letter. I especially would like to see your response to Reviewer 1’s comments about a lack of convincing evidence for swimming preadaptation. In this light, it might be prudent for your revision to admit speculation whenever possible and place your entire study in more of a hypothetical context.

There are many minor suggestions to improve the manuscript. Note that reviewer 2 kindly provided a marked-up version of your manuscript.

Therefore, I am recommending that you revise your manuscript, accordingly, taking into account all of the issues raised by the reviewers.

Good luck with your revision,

-joe

Reviewer 1 ·

Basic reporting

I admit this has been a bit difficult ms to revise as there are overall some meaningful aspects brought to light. However, focusing on its logical structure and theoretical background, I have to highlight some formal and conceptual flaws that lead me to suggest for a full rejection because some decisive steps to streamline methodology with the premise and to bring interpretations in line with evolutionary thinking are lacking.

The ‘story’ starts from the empirical observation that if some fully terrestrial caterpillars happen to fall by chance into water, they are capable to “swim” on the surface, then to escape predation and safely reach the nearby vegetation.

The methodology then involves studies on such kind of locomotion by throwing into water caterpillars of several terrestrial species and recording those capable of “swimming”, either via undulatory or kick movements.

The analysis details aspects of such locomotion according to larval instars, body mass, shape etc.

The conclusion claims that such “swimming” attitude is a preadaptation which may explain colonization of aquatic environments by caterpillars not just of Noctuoidea but in general.

Granted that there are some known swimming caterpillars in Noctuoidea that regularly inhabit semi-aquatic or regularly flooded environments, above all those of the Neotropical genus Paracles (duly recorded in the ms), in this piece of research I see no relation between the main assertion in the premises, that of some typically terrestrial caterpillars can swim, and the methodology to proof such assertion. In fact, the methods just detail aspects of the locomotion giving for granted that this is a kind of swimming, but real swimming has yet to be demonstrated. Hence there is a logical leap in how reasoning develops.

I see an obvious difference between normal locomotory reactions shown by caterpillars placed under stressful circumstances and the proper concept of “swimming”. For instance, when they face a menace, many species show the so-called jerking behavior (see Stamp & Casey 1993. Caterpillars), which seems to exactly correspond to the undulatory movements seen in the videos accompanying the ms. The only difference would be that with normal jerking the larva holds onto the substrate with the false-legs, while in liquid this cannot happen and the undulations thus propagate towards the end of the body. Whilst the looping or semi-looping locomotion seems at the base of the kick movements seen in another video. I suspect then that there are much more parsimonious explanations accounting for such kinds of movements in the aquatic medium.

Experimental design

As just said, I see no links between the driving idea of the ms and methods. They are totally uncoupled.

I would like to see a methodology oriented towards probation of the initial really heavy assertion that such terrestrial species do swim, otherwise this would only remain postulated. I would like to see a kinematic comparison among exclusively terrestrial Noctuoidea caterpillars, supposedly/occasionally swimming ones and truly swimming ones, ideally also with anatomical analysis of their muscles.

Validity of the findings

Due to failure in probating the suggested evolutionary path, any assertion about preadaptations has to be challenged too. I think no preadaptation can be claimed to have been discovered unless it has been realized. Only when one adaptation has been achieved, some premise may be inferred to have represented a preadaptation.

However, let’s admit that the research unveiled a basic locomotory pattern which may explain swimming in other groups.

It becomes essential here to bring some phylogenetical thinking into the analysis, otherwise all variables and players would remain loose and ultimately unrelated. For sure there are no direct relationships between arctiine Paracles and the species of present trials, thus how can the latter reveal a preadaptation realized in the former? In theory it could be done, but the authors should be able to demonstrate that there is a common groundplan locomotory attitude shared by a monophyletic group of Noctuoidea which subsequently diversified into swimming by some lineages and jerking (or kicking) by others (or both).

In the ms instead, the conclusions are pushed even further, being extended not just to Noctuoidea, something that has to be demonstrated yet, but in general to aquatic caterpillars.

Very weak are also claims that six of the seven "swimming" species occur in riparian habitats and make use of such preadaptation to rescue themselves, thus implicitly assuming that it is an adaptation proper. This is fully speculative and subjective, in the absence of a thorough analysis of the ecological distribution of such species, some of which (Spirama retorta, Hypopyra vespertilio, Dinumma deponens and Lymantria dispar) I can confirm are non-hygrophilous elements. Laelia coenosa, Naranga aenescens and Acosmetia biguttula are instead hygrophilous. So what is the relationships among them? How a same explanation can be extended to species with such different environmental requirements? Are the authors assuming that the "preadaptation" works already as an "adaptation" in some species? How all this stands together given their phylogenetic separation? Only explanation is that the attitude is homoplasous, but then it cannot be a synapomorphy, then there would be no shared preadaptation.

Additional comments

To sum up, I think the core part with the analysis of locomotion is meaningful but it does not really fit with the main scope of the research. At the present state of knowledge, the ‘pseudo-swimming’ shown by some terminal taxa seems more parsimoniously ascribable to known locomotory patterns expressed under unusually stressful circumstances. It should not be claimed to reveal a preadaptation to ‘swimming’ in other far apart, not directly related terminal taxa.

I think also the authors may have really opened a window on some ancestral locomotory character and encourage them to carry on with their research born after a serendipitous observation, but everything should be demonstrated, while as of now, the work essentially suggests an analogy, which seems a too weak justification for deserving publication. Showing that such “analogy” is a homology would instead decidedly convert the work from exclusively speculative to a sound scientific one.

Speculation is welcome in the journal, but this should at least build on some achieved results, while at this state of the study I see admittedly none.

·

Basic reporting

This strikes me as a tight, careful study, and fully worthy of publication in PeerJ. The manuscript is concise, well edited, focused, well organized, and well referenced. The amount of information in the Table 1 is impressive. The finding that swimming ability increased with size (across instars) is novel and well documented. My comments are minor and mostly represent suggestions, rather requirements.

Experimental design

Very good. Experimental design, details of methods, statistical analysis, and figures appear to be strong.

Validity of the findings

See below. I have only minor quibbles.

Additional comments

The abstract should be recrafted to down-weight the mention of the evolutionary/comparative aspects of the study, which get little treatment, and are not (yet) carried out or discussed in a phylogenetically (rigorous) manner—see below. This sentence could be deleted or re-worded: “Because some aquatic caterpillar species belong to these moth families, we posit that the active swimming observed in these terrestrial caterpillars represents preadaptation to an aquatic environment.” (It’s fine to speculate about these matters in the Discussion. See also below.) To the Abstract they should make mention of some of their key experimental results, which are not yet mentioned, e.g., that the frequency of active swimming behaviors and speed of swimming increased across instars/body size, and perhaps other key findings.

Some mention of surface tension of water and its role in the observed floating and swimming is warranted, especially because there are caterpillars that swim below the surface of the water. This matter might be handled either in the Introduction or Methods. Somewhere they should state that they only dealt with taxa that swim at the surface of water (aided by water tension). (And while the physical properties of water tension are recognized as important, they are not treated in the study.) I wonder if the authors observed any species that broke through the water surface? What happened?

Table 3 and 4 appear unnecessary and that the essential data therein could simply be added to text.

The evolutionary arguments are a bit weak. While the authors claim that the observed swimming behaviors across four families represent a pre-adaptation, and they are probably correct, they did not carry out comparative tests, e.g., of whether related caterpillars of arid environments also had innate swimming behaviors. Or make comparisons between surface swimmers and fully aquatic taxa. No mention is even made of the locomotory movements of fully aquatic species. In a concluding sentence the authors state, “Active swimming will likely be found in other lepidopteran families (e.g., Crambidae) that include aquatic caterpillar species.” Which, to my mind, argues that swimming behaviors are selected for/under selection and thus seemingly (weakly) argues against swimming being a pre-adaptation. Regardless, I would tighten up the logic and language; perhaps simply suggest in the Discussion that the elongate bodies and undulating, locomotory actions of caterpillars on land appear to have pre-adapted Lepidoptera for active swimming behaviors. And these behaviors in turn, have pre-adapted some caterpillars to become fully aquatic.

Unless the arguments can be bolstered/expanded in the Discussion, I think the evolutionary claim needs to be downweighted in the Abstract. If it is retained in the Abstract the authors should give the basis for the conclusion, something akin to: The discovery of active swimming behaviors across four different families of Lepidoptera is suggestive that the elongate bodies and locomotory systems of moth larvae have pre-adapted the lineage for swimming.

I suggested edits and added these to a pdf version of the manuscript that I am returning.

In sum, I think this a strong effort that is novel, well documented, handsomely illustrated, and acceptable for publication with only minor revisions.

David Wagner

PS As a side note, I have observed woolly bear caterpillars (Arctiinae) swimming, and foraging underwater in the Neotropics—craziest thing I have ever seen! I think the caterpillars that I observed were members of the genus Paracles. I reared one in aquarium on submerged lettuce and lawn weeds! The fully fed larva came to the surface, formed a floating mat out of its own setae that it tore off its body with its mandibles, and then pupated on the floating raft of setae!

---

## Round 0.2 · Minor Revisions

Dear Drs. Hayashi and Sugiura:

Thanks for revising your manuscript. The one reviewer willing to re-review is mostly satisfied with your revision (as am I). Great! However, there are a few minor issues to entertain. Please address these ASAP so we may move towards acceptance of your work.

Best,

-joe

Reviewer 1 ·

Basic reporting

Having reviewed a previous version of this ms, I will be now very straightforward and state that I am very happy with the current draft. The findings are interesting, worth of being shared with the scientific community, all unsubstantiated speculations not even fitting with current knowledge on both ecology and systematics of the families involved have been removed, and from now on further research may be build upon the current findings.

Experimental design

All is clearly stated and replicable. I may just stress that is more usual to see the three dimensions of the plastic 'vessel' (possibly better 'container') in mm and not as mm3, as only after the multiplication is done the mm3 are got.

Validity of the findings

Ok.

Additional comments

I am recommending a truly minor revision to just adapt to the following suggestions:
1) per instar stage - looks redundant, as the stage is the larval one. only 'per instar' would be ok;
2) despite it can be used after the plural, behaviour is usually used as singular, especially in sentences like 'two types of behaviour', but also on other occasions. I guess it may be mostly employed at the singular throughout the ms;
3) last sentence of animal ethics - it should be areas;
4) ref Sugiura S, Takanashi etc. should be moved above Sugiura S, Yamazaki K.
5) see above for mm, mm3;
6) I would not say 'silk threads produced from spinnerets' as silk is produced by the relative glands whose secretion is emitted through the spinneret; better to use 'emitted', 'released' or some equivalent expression;
7) in using the silk threads for hanging down by the caterpillars, the important phenomenon of 'ballooning' to disperse at the larval stage has not been mentioned at all;
8) the ref 'van Griethuijsen LI, Trimmer BA. 2014' should better be placed as 'Griethuijsen LI, van, Trimmer BA. 2014' due to small 'v', also because that's the way after it is cited in the main text;
9) In the discussion, the sentence that 'Aquatic behaviours have been reported in some aquatic and semi-aquatic caterpillars' looks obviously circular. Furthermore, in the examples that follow it may look weird to many readers that Ostrinia penitalis is mentioned while the important aquatic group of other pyraloids such as Acentropinae/Nymphulinae (or Nymphulini) is omitted.

·

Basic reporting

See below

Experimental design

See below

Validity of the findings

See below

Additional comments

This paper is original, includes novel observational from field and lab. It is focused, grounded, well-written. The figures and videos are great. Overall, I find the manuscript acceptable with minor revision.

There is substantial redundancy that can be eliminated—for example, I recall some six places where the authors mention that there are seven species that exhibit “forward movement on the water.” Geesh. The same ideas get repeated too many times. I recommend trying to shorten the Abstract and text by 10%.

The hypothesis in the introduction is forced, uninteresting, and unnecessary as written. Why does one need a hypothesis, “that some terrestrial caterpillars can exhibit forward movement on the water surface” when you already observed the behavior and filmed it? It’s just filler, if this is going to be the hypothesis. This is largely a novel observational study—it doesn’t demand a hypothesis.

I disagree with other reviewer that this is not swimming. According to Merriam Webster:
Definition of swim; intransitive verb”:
1a: to propel oneself in water by natural means (such as movements of the limbs, fins, or tail)
b: to play in the water (as at a beach or swimming pool)
2: to move with a motion like that of swimming : GLIDEa cloud swam slowly across the moon
3a: to float on a liquid : not sink

The text would read better if the action were called swimming. Other workers will NOT be able to find the work, if it is called something other than swimming. No one that is interested in swimming will be able to search and find “forward movement on the water surface.”

The larger matter is why they “swim” and I think this was the issue the other reviewer had with the first draft. Intent. We/you don’t know why they exhibit swimming behavior. The behavior could be multifunctional. Beyond helping to avoid predation, the behavior is necessary to reunite the caterpillar with its hostplant, it also prevents drowning, and may return the larva to substrate upon which it could/would settle until the time to feed again. If you place an erebid caterpillar on glass over white paper, many would move to a new site of safety where their coloration would more cryptic. Caterpillars see. They assess their light environments. In sum, I would call it swimming, but think you should backpedal on why they swim—that is unknown.

Please discuss/mention the two families that did not exhibiting swimming behaviors—their failure to swim is important.

The larvae does not kick the anal prolegs to swim. Look at your videos. The caterpillars use the entire abdominal terminus, folding it under and then flicking it rearward. No less than segments A7-A10 are involved. This is flicking behavior, not kicking. This is more analogous to the snapping movements of fleas and Mexican jumping bean caterpillars than it is when kicking, i.e., where the appendages propel the individual. It is not definitely an error to state anywhere in the manuscript that these insects are kicking with their anal prolegs. Categorically false. Nor is the movement equivalent to the smooth upward and downward movements of cetacean swimming propelled by flukes. I am surprised I didn’t catch this the first time I read this paper—and apologize. This must be corrected before I would recommend the manuscript for publication.

I made a Word version from the pdf, and added many edits to text and figure legends. Many of the suggestions will help to make the text more concise and less redundant. Happy to help with a last read before resubmission or help in other ways.

---

## Round 0.3 · Major Revisions

Dear Drs. Hayashi and Sugiura:

Thanks for revising your manuscript. Both original reviewers have now commented on your revision, and both believe your emphasis on swimming is just too much. I agree with them. I am offering you the chance to revise once more, but you also may wish to submit the article in its current format and theme to another journal. The speculative nature of this work and the lack of support for such behavior in these caterpillars, cannot be published in the manuscript’s current form.

If you choose to revise your work, I am recommending that you seriously take into account all of the issues raised by both reviewers.

Best,

-joe

Reviewer 1 ·

Basic reporting

Being already a third review round, please go directly to final commenting.

Experimental design

Ditto.

Validity of the findings

Ditto.

Additional comments

Unfortunately, the authors have taken the opportunity of a comment by the other reviewer that the locomotion observed may be termed “swimming”, and turned back to their original interpretation, which undoubtedly would make the title catchier. I am again here only to avoid that other scholars would make fun of the authors like it happened at a congress presentation that I attended many years ago. The presenter reported on the “swimming” behavior by terrestrial ground beetles (Carabidae) (note: terrestrial like the caterpillars of ours) that had been thrown in a pool to observe their “escape” strategy (note: chance events like those of ours in an overwhelmingly hygrophilous group). The audience burst into hearty laughter and sometimes, when participants to that congress meet, they still recall such episode with great fun despite over 30 years have elapsed. “Do you recall the silly talk of the swimming carabids?”

This work is perfectly twin to that research and I would not do a good job if I recommended to call the behavior as swimming. Authors: do you prefer a more fashionable title at the risk of damaging your reputation or would you just be satisfied with a nice article published in PeerJ avoiding any hindrance to your career? I would have no hesitation if I had to recommend a choice to my students.

Regarding terminology, it is a well-known fact that many terms in life sciences are borrowed from common speech, but they regularly have a more restricted meaning. In case of interest, there is only one word whose meaning is broader in biology (anthropology) than in common speech, that is “culture”. Accordingly, despite dictionary definitions, “swimming” in biology assumes a kind of locomotion made possible by a coordinated integration of muscles, nerves, proprioceptors and stimuli that could only have developed as an evolutionary response to some ecological needs (that’s why the original idea of “preadaptation” was unacceptable). These caterpillars are not aquatic, they fall in water by chance and at a ratio impossible to drive an evolutionary path, let alone the maintenance of such a complex anatomo-physio-ethological system as if real natatory behavior was involved.

If the authors want to pursue with their intent, I do not want to feel responsible for their reputational damage and am compelled to recommend again a full rejection.

Nothing in biology makes sense except in the light of evolution (Dobzhansky, 1973).

·

Basic reporting

This is a clear, significantly shorter version, without much of the redundancy of the previous drafts. The authors have incorporated nearly all my suggestions from my two previous reviews, including my line edits to their text. So I have only a couple minor reservations.

The new paragraph on the evolution of the swimming in (terrestrial) caterpillars is speculative and based on extremely small sample sizes, but it is so short that I feel it is fine. But in three separate sentences, the authors make a sweeping statement that surely has exceptions. All three sentences require qualifying language--I have made some suggestions that the authors may find helpful. I have also offered suggestions to the Conclusion.

Experimental design

NA

Validity of the findings

The findings are clear, succinctly presented. The videos provide a great rendering of the swimming behaviors

Additional comments

This is a clear, significantly shorter version, without much of the redundancy of the previous drafts. The authors have incorporated nearly all my suggestions from my two previous reviews, including my line edits to their text. So I have only a couple minor reservations.

The new paragraph on the evolution of the swimming in (terrestrial) caterpillars is speculative and based on extremely small sample sizes, but it is so short that I feel it is fine. But in three separate sentences, the authors make a sweeping statement that surely has exceptions. All three sentences require qualifying language--I have made some suggestions that the authors may find helpful. I have also offered suggestions to the Conclusion.

---

## Round 0.4 · accepted · Accept

Dear Drs. Hayashi and Sugiura:

Thanks for revising your manuscript based on the concerns raised by the reviewers. I do appreciate your persistence with this review process. I now believe that your manuscript is suitable for publication. Congratulations! The videos are exquisite! I look forward to seeing this work in print, and I anticipate it being an enlightening resource for groups studying swimming-like behaviors in insects. Thanks again for choosing PeerJ to publish such important work.

Best,

-joe